# ‘Dr. Google, What Is That on My Skin?’—Internet Searches Related to Skin Problems: Google Trends Data from 2004 to 2019

**DOI:** 10.3390/ijerph18052541

**Published:** 2021-03-04

**Authors:** Mikołaj Kamiński, Linda Tizek, Alexander Zink

**Affiliations:** 1Individual Medical Practice, Bogdanowo, 64-600 Oborniki, Poland; 2Department of Dermatology and Allergy, School of Medicine, Technical University of Munich, 80802 Munich, Germany; linda.tizek@tum.de (L.T.); alexander.zink@tum.de (A.Z.)

**Keywords:** Google Trends, Internet, skin manifestations, perspiration, pruritus, infodemiology

## Abstract

The Internet is a common source of health information as search engines propose websites that should answer users’ queries. The study aimed to investigate the search behavior of Google users related to skin clinical signs as well as to analyze their geographical, secular, and seasonal patterns. The data of Google Trends was used to analyze the number of Google searches related to skin problems from January 2004 to December 2019. Thirty-four topics representing dermatologic complaints were identified. The interests of all topics were calculated in proportion to the Relative Search Volume (RSV) of ‘Scar’. Geographical patterns as well as secular and seasonal trends were analyzed. Countries with few users who searched for skin problems were excluded from the analysis. Globally, gaining the most attention were ‘Itch’ proportion to RSV of ‘Itch’ (2.21), ‘Hair loss’ (1.56), ‘Skin rash’ (1.38), ‘Perspiration’ (1.32), and ‘Scar’ (1.00). In 42 of the 65 analyzed countries, ‘Itch’ was the most popular topic, followed by ‘Hair loss’ (*n* = 7), and ‘Pustule’ (*n* = 6). The RSV of all topics increases over time, with ‘Comedo’ (5.15 RSV/year), ‘Itch’ (4.83 RSV/year), and ‘Dandruff’ (4.66 RSV/year) being the most dynamic ones. For 23 topics, the highest interest was noted during warm months. Considering skin manifestations, Google users are mainly interested in itch, hair loss, and skin rash. An increasing number of individuals worldwide seem to use Google as a source of health information for dermatological clinical signs during the study period.

## 1. Introduction

Skin diseases are one of the most common health problems, affecting up to one-third of the general population worldwide [1]. In the Global Burden of Disease Study, skin diseases were reported to be responsible for 1.8% of the global burden of all human diseases [2], with three skin conditions (fungal skin diseases, acne vulgaris, and other skin and subcutaneous diseases) being among the top ten of the most prevalent diseases [1]. Dermatologic diseases decrease quality of life [3,4] and some patients with a severe course of a skin disease have to consult many specialist or complementary medicine practitioners until they find relief [5].

The Internet has become an essential source of health-related information as the number of search queries has been rising continuously for many years [6,7,8,9]. Not only for people who have some symptoms and want to inform themselves prior to consulting a physician, but also for patients seeking for further information like treatment options [10]. In France, for example, 80% of young adults perceive the Internet as a reliable source of health information [11]. When searching for health-related information on the Internet, there is a large amount of information on websites or social media platforms that allow people to share information with other affected people [12,13,14,15]. However, some studies reported that there are some websites where the information regarding dermatologic conditions is of low-to-moderate quality, which could be misleading for people [16,17].

Since not all people with clinical skin signs consult a physician directly [18], using data provided by search engines like Google might help to assess people’s interest in different clinical skin signs and, therefore, help to improve a more people-centered care. Google is the most popular search engine across the globe [19]. Previous studies using data on Google already demonstrated that analyzing Google search data are useful to identify specific interests or medical needs like studies demonstrated that there was a surprisingly high interest in anal pruritus [6,20]. Zink et al. utilized Google Ads Keyword Planner to analyze interest over time in pruritus in Germany and to establish the most prevalent body location of pruritus [6]. Wongvibulsin, inspired by the first paper, analyzed pruritus location for citizens of the United States [20]. Moreover, seasonal variations in the number of Google searches related to specific diseases such as skin cancer were observed which could also help to detect the burden of various diseases outside the medical setting [21,22,23].

There are a few studies using Google data for one specific clinical skin sign in one country [6,20], but there is no study that investigates numerous clinical skin signs together in several countries using Google Trends. Therefore, this study aimed to examine several skin problems in various regions to identify which clinical signs are the most distressing ones based on their number of Google searches to identify geographical differences and seasonal variations.

## 2. Materials and Methods

### 2.1. Data Collection

This is a retrospective study using freely available data provided by Google Trends (GT) and, thus, no ethical approval was necessary (https://trends.google.com/trends/ (accessed on 11 February 2021).

As shown in previous studies [7,24,25], GT allows the analysis of relative search volume (RSV) of search terms in the Google search engine. RSV ranges from 0 to 100, with 100 corresponding to the peak of popularity (100% of popularity in given period and location) and 0 complete lack of interest (0%) [26]. The tool enables to compare a chosen term in a specific region and time since January 2004. The comparison could involve up to five terms at once. When comparing multiple search terms, RSV is adjusted and RSV = 100 represents the highest popularity of one of the chosen phrases.

GT may recognize the input as ‘search term’ or ‘topic’. Search terms are the exact words that were searched for, while topics could include words that are proposed by the engine when it recognizes phrases related to the popular query. In contrast to search terms, topics enable to compare the given terms between all countries. For example, the search term ‘mouse’ will be analyzed by GT literally; thus, RSV will be the highest in English-speaking countries, while the topic ‘mouse’ will include all queries associated with the topic in all available languages.

The data from 1 January 2004 to 31 December 2019 were extracted from GT. The book ‘Clinical Dermatology’ by Carol Soutor was used to create an initial list of clinical skin signs [27]. Then, the identified clinical skin signs were typed into GT to check whether the input matched any topic. Thus, a total of 34 topics related to dermatologic clinical signs were identified: ‘Abrasion‘, ‘Blister’, ‘café au lait spots’, ‘Cellulite’, ‘Comedo’, ‘Dandruff’, ‘Eczema’, ‘Erythema’, ‘Eschar’, ‘Freckle’, ‘Hair loss’, ‘Hyperpigmentation’, ‘Hives’, ‘Itch’, ‘Liver spots’, ‘Melanocytic nevus’, ‘Melasma’, ‘Nevus’, ‘Nodule’, ‘Papilloma’, ‘Papule’, ‘Perspiration’, ‘Petechia’, ‘Pustule’, ‘Scar’, ‘Skin fissure’ ‘Skin rash’, ‘Skin tag’, ‘Skin ulcer’, ‘Stretch marks’, ‘Telangiectasia’, ‘Vesicle’, ‘Wart’, and ‘Xeroderma’. The names of many topics are professional, but many common terms such as ‘mole’ matched one of the analyzed topics, ‘Melanocytic nevus’; ‘blackhead’ matched ‘Comedo’, ‘sweating’ matched ‘Perspiration’, ‘dry skin’ matched ‘Xeroderma’, etc. In the analysis, matched topics representing dermatological diseases such as ‘Psoriasis’ were not considered. No topic matching bulla, crust, lichenification, skin atrophy, or skin scale was identified. All chosen topics were typed into GT separately (data generated for only one topic is later called as non-adjusted data) and were compared with the topic ‘Scar’ (data generated for two topics at once is called adjusted data). Only two topics were compared at once. The topic ‘Scar’ was chosen as references because in most of the 65 countries it was the topic with the highest number of low search volume. Therefore, ‘Scar’ enables analysis of the popularity of topics in all included countries. In general, data was planned to collect from each country in the world, but GT automatically excluded regions with low search volume since searches were made by an unrepresentative group of Google users and, thus, the data would be susceptible to irregular variations. According to a previous protocol, details on search conditions and inputs are reported in detail (Appendix A) [26].

### 2.2. Data Processing and Statistical Analysis

The RSV was set to ‘0.5’ by all data points described by GT as ‘<1%’ and ‘0.1’ to all RSV data points equals ‘0%’. To calculate the proportion of adjusted RSV of all topics, the topics were adjusted to the topic ‘Scar’, which as reference had an RSV of ‘1.00’ (Appendix A). We reported RSV as index (from 0 to 100) and as percentage (from 0 to 100%).

The adjusted data of compared break by region represents the proportion between RSV of topics and ‘Scar’ in a specific country (Appendix A). The sum of RSVs of both topics (benchmark ‘Scar’ and one of the 33) in a given region equals 100. This allows to analyze which searches were more often searched for in each country. We calculated the most frequent dermatologic problems-related topics for all countries with significant search volume. Because the proportion of the popularity of the topics was always adjusted to the topic ‘Scar’, the RSV of ‘Scar’ was set to 50 in all regions.

We used non-adjusted data to establish countries with the highest RSV for each topic (Appendix A). In this analysis, RSV equal to 100 represents the country with the highest number of queries for each topic adjusted to the number of Google users in the region. Therefore, the rank of the countries represents activity of the Google users in a country, not crude search volume. Countries with less than five topics with RSV above null were excluded from the analysis.

Non-adjusted data were used to perform time series analysis (Appendix A). The Seasonal Mann-Kendall test was performed using R 3.6.1 (R Foundation, Vienna, Austria) and the *Kendall* package to assess the presence of a significant secular trend of time series [28]. *p*-Value < 0.05 was considered as a significant difference. For all significant secular trends, a univariate linear regression was performed to estimate slope expressed as changes of RSV per year. To analyze seasonal variation, an exponential smoothing state-space model with Box-Cox transformation was fitted, autoregressive-moving average errors, trend, and seasonal components (TBATS) using the *forecast* package of R to the time trend [29]. We extracted the seasonal component of time series using the Seasonal Decomposition of Time Series by Loess (Local Polynomial Regression Fitting). We calculated the yearly amplitude of the seasonal component of the time series by subtracting the maximum seasonal component from the minimal component.

We performed a sensitivity analysis because most of the countries considered are in the Northern hemisphere. Thus, we generated non-adjusted data on interest over time for all topics for three southern countries: Brazil, South Africa, and Australia. We repeated seasonality analysis for these countries.

Furthermore, we generated GT data (adjusted and non-adjusted data on interest over time) for topics representing skin diseases: ‘Atopic dermatitis‘, ‘Basal-cell carcinoma‘, ‘Melanoma‘, ‘Psoriasis‘, ‘Rosacea‘, ‘Scabies‘, and ‘Squamous cell skin cancer‘. Moreover, we generated similar data for non-medical topics: ‘Car‘, ‘FC Bayern Munich’ (world-class football team), ‘Rome‘, ‘Star Wars‘, and ‘Tomato‘. This sensitivity analysis aimed to compare the relative interest of several skin diseases and non-medical topics in comparison to ‘Scar’ as well as to analyze the time trend. This analysis was performed to investigate whether the popularity of skin manifestation was similar to the interest in skin diseases or non-medical topics and to test whether there was a general increase in popularity due to the increase of Google users in the analyzed period.

The dataset will be available on Mendeley after publication of the paper. The data collection and data processing flowchart is presented in Appendix A.

## 3. Results

From January 2004 to December 2019, ‘Itch’ (2.21), ‘Hair loss’ (1.56), ‘Skin rash’ (1.38), ‘Perspiration’ (1.32), and ‘Scar’ (1.00) had the highest overall interest of the 34 analyzed clinical skin signs (Table 1). In Figure 1 and Figure 2, we visualized the most frequently searched topics related to clinical skin signs in each of the 65 included countries. The topic ‘Itch’ was most frequently searched for in 48 countries, whereas ‘Hair loss’ was most common in seven countries. The five most common topics in each country are represented in Appendix A. Moreover, we presented five countries with the highest non-adjusted RSV by region for all topics in Appendix A. ‘Cellulite’ was particularly popular in countries of South Europe. ‘Comedo’, ‘Hair loss’, and ‘Scar’ were of great interest by people living in South East Asia countries. Queries associated with ‘Wart’ were generated mainly by users from Balkans. ‘Pruritus’ generated the highest interest on Caribbean Islands.

There was a significant increase in the interest of all topics during the study period (Table 2, Figure 3). The RSV most dynamically increased over time for topics ‘Comedo’ (5.15 RSV/year), ‘Itch’ (4.83 RSV/year), and ‘Dandruff’ (4.66 RSV/year). Moreover, except for ‘Hair loss’, seasonal variations were observed. We observed the highest RSV in July (*n* = 16 topics), June (*n* = 5 topics), and August (*n* = 2 topics). The lowest interest was noted in December for 24 topics. The highest yearly amplitude had the following topics: ‘Cellulite’ (43.77 RSV), ‘Liver spot’ (33.35 RSV), and ‘Vesicle’ (31.49 RSV).

In the sensitivity analysis, we found that most of the seasonal variations of the analyzed topics realted to skin problems were opposite to that observed in the main analysis for the world (Appendix A). The interest in non-medical topics was higher than the reference topic ‘Scar’ (>1.00), while the interest in topics representing skin diseases was lower than ‘Scar’ (<1.00) (Appendix A). The interest in ‘Atopic dermatitis‘, ‘Psoriasis‘, and ‘Scabies‘ increased, while it decreased in ‘Melanoma‘ (Appendix A and Appendix A). Comparable to skin problems, all topics regarding skin diseases as well as non-medical topics revealed seasonal variation (Appendix A and Appendix A).

## 4. Discussion

The aim of the study was to examine whether there are differences in the interest regarding clinical skin signs in various countries around the globe. We found that the interest in skin manifestation increases since 2004 and reveals seasonal and regional variations.

The Internet has become a common source to search for health-related information. In general, there might be various reasons why in some regions there is a higher popularity rank such as (i) a higher prevalence of the underlying condition, (ii) a higher subjective discomfort, and (iii) that for some clinical signs there are more opportunities for efficient self-treatment. Overall, ‘Itch’, ‘Hair loss’, ‘Skin rash’ and ‘Perspiration’ gained the most attention, which may reflect the real-world relative prevalence of these clinical signs or the burden of unmet health needs. Indeed, pruritus as skin disease affects around 4% of the global population [1]; however, it is also a very common symptom of dermatitis, scabies, fungal skin diseases, and urticaria, which were the fourth most common skin diseases according to the Global Burden of Disease Study in 2013 [2]. Hyperhidrosis bothers 4.8–16.3% of adults [30,31], but our Google analysis may suggest that less common pruritus has more troublesome clinical signs than excessive perspiration. Three of the top five topics (hair loss, perspiration, and scar) are rather cosmetic concerns than signs of severe disease. Similarly, the melanocytic changes that may include melanoma were beyond the top ten topics. Interestingly, it is postulated that the majority of the adults may have dandruff [32], but the rank of the topic does not reflect this high prevalence. The skin manifestations that caused burdensome suffering (e.g., the localization of the changes is hard to cover, or the pruritus is persistent) are among the top topics: pruritus, perspiration, skin rash, and hair loss. Conditions such as dandruff and abrasion are common, but they can be treated without consulting a physician; thus, they might not require persistent searching for a treatment. Inversely, the persistent pruritus or hair loss may be challenging for treatment even for medical specialists. Nevertheless, the proposed theoretical framework may explain general tendencies that should be verified in further studies.

The data suggested that the RSV of all topics increased over time, which may be explained by an increase of the search engine users. It might also indicate that there was also an increase in the number of search queries as the Internet is becoming a more common source for health-related information. Since there are many options for self-treatment of specific skin conditions, people might search online for required medications. The most dynamically grown searches were on comedo, pustules, and pruritus, which is consistent with the surging prevalence of acne vulgaris (comedo) [33,34] and findings of the Global Burden of Disease Study (pruritus, pustules) [1]. The fourth topic that most dynamically gained attention was melasma. The condition affects approximately 1% of the general population and 9–50% of the high-risk population of melasma such as pregnant women and those using female hormones [35,36]. We assume that the observed search trends may be caused by broader recognition of melasma as treatable lesions. Moreover, we found that interest in some non-medical topics and topics representing skin diseases did not increase over time. This suggests that the increase in the number of Google users over time is not always related to the rise of RSV of all topics.

Most of the topics revealed seasonal variation. Since most of the Google users live in the Northern Hemisphere, the global trends mirror seasons of this part of the earth. In several conditions, the seasonal variation confirms previously detected patterns: pruritus [6,37], dandruff [38], and xerosis [39,40] had the highest RSV in winter, atopic eczema [41] in spring, perspiration [27] during summer, hair loss pattern in late summer/autumn [42], and leg ulcers in autumn [43]. Abrasion and feet blisters may occur more commonly in warm months due to outdoor activities. Lesions rich in melanocytes may increase activity during the period of increased insolation. The peak of interest in erythema in July may be associated with more frequent sunburns and insect bites. One of the causes of petechiae might be Henoch-Schönlein purpura. The disease tends to be most commonly diagnosed in children in spring, but among adolescents in summer [44], which is similar to the observed seasonal trends. The prevalence of urticaria might be positively associated with temperature [45]. This relationship may explain the peak of interest in the hive during July. Interestingly, aggravation of acne is most commonly observed in winter [46,47]. However, the search volume on comedo or pustules-related information peaks in July. The increased interest in many topics (e.g., cellulite, comedones, melanocytic lesions, papules, pustules, skin tags, stretch marks, etc.) during warm months may be caused by the desire to remove unpleasant lesions to comfortably expose the body during summer activities. In most of the cases, the interest was the lowest during December. It was previously described that the activity of Google users might decrease during the Christmas holidays [24]. In other cases, the nadir of RSV may be related to the seasonality of the underlying conditions. Interestingly, hair loss did not reveal significant seasonal variation. Previously, Hsiang et al. found that interest in hair loss among Google users in English-speaking countries peaks in summer and autumn [48]. We found that exclusion of the unusual peak of interest (February 2007) allowed to detect seasonality with the highest interest in August, and the lowest in February. However, we cannot find news that may explain this peak of search volume. Our sensitivity analysis suggested that southern countries may have opposite seasonal variations than that observed in the main analysis for the whole world. However, in many cases (especially for South Africa), the significant seasonal variation was not observed, which may be caused by the limited number of searches, which may be more susceptible to the irregular variations and modification of interest by media clamor.

Internet-derived data creates unconventional opportunities for medical researches. The analysis of Google queries may help to detect unmet health needs in a population, which otherwise would have to require extensive epidemiological studies [6,20]. A preliminary analysis of GT may be useful to create a background for future studies because this process is quick and costless. In this study, we rank the skin-related complaints using freely accessible data. Importantly, we found that the interest in all of the skin problems increases over time. The increased interest of users regarding skin complaint warrants a call to action. Firstly, the quality of websites has to be improved as many websites have a low-to-moderate quality [16,17]; thus, professionals should recommend the Web communities using reliable websites. Secondly, previous studies showed that online forums for individuals with skin diseases might contain misleading content on treatment [49,50]. Therefore, the public should be aware of this danger. Finally, the implementation of AI-based self-diagnosis tool might help to deal with the growing interest of Google users in skin conditions. Currently, the abilities of apps for skin lesions recognition are limited [51], but it may be expected that future generation of AI tools may be helpful for quick differentiation, which changes require consultation with dermatologists.

### Limitations

This study has some limitations. Firstly, the popularity of Google differs across the globe. For example, the search engine is less prevalent in Russia (approximately 50% of the Internet users), China (currently less than 5%), and the United States of America (80–85%) than in the European Union (over 90%) [19]. Therefore, the results might be more representative for regions with a higher market share of Google. Secondly, the GT does not provide the exact number of searches; thus, we ranked the topics using the relative index. Thirdly, the data does not provide information regarding users’ age, gender, and other searches. In consequence, it is not possible to draw any conclusions about people’s characteristics that generate queries. We may expect that Google users may be younger on average than the general population, which may explain the high interest in perspiration and pustules. Fourthly, the study does not explain the causes of the increase of Google users’ interest in clinical skin signs. People may get used to searching for information on their health issues on the Internet and we do not know how their search behavior is affected by targeted e-marketing. Finally, the results of Google data analysis should be considered carefully: the statistical tests may be susceptible to an irregular pattern and do not detect significant trends.

## 5. Conclusions

Globally, Google users showed the highest interest in itch, hair loss, and skin rash. GT may be a feasible tool for the assessment of time and geographical patterns of different skin manifestation. Google is gaining popularity as a source to search for information on dermatological problems. Professionals should recommend reliable websites for their patients.

## Figures and Tables

**Figure 1 ijerph-18-02541-f001:**
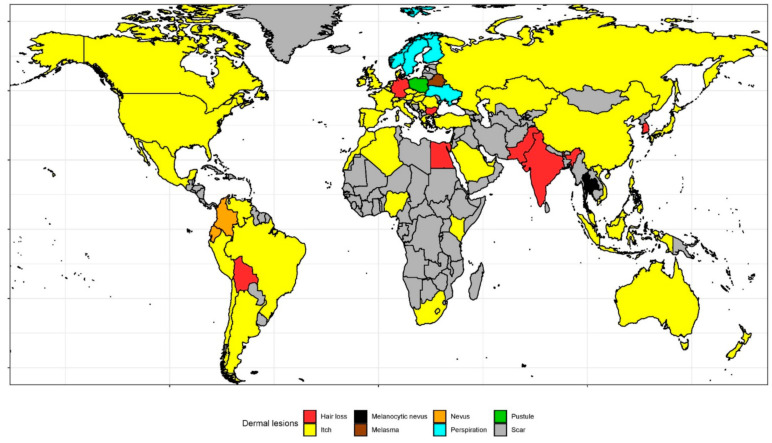
World map of the most popular topic representing skin problem in each country.

**Figure 2 ijerph-18-02541-f002:**
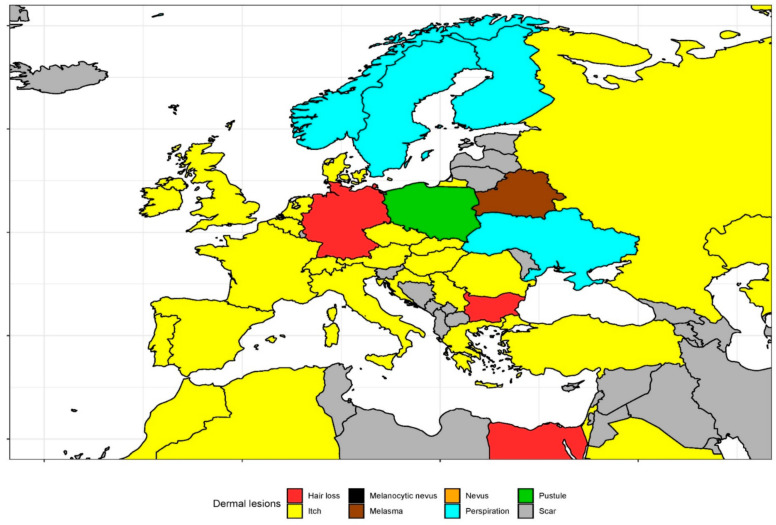
Europe map of the most popular topic representing skin problem in each country.

**Figure 3 ijerph-18-02541-f003:**
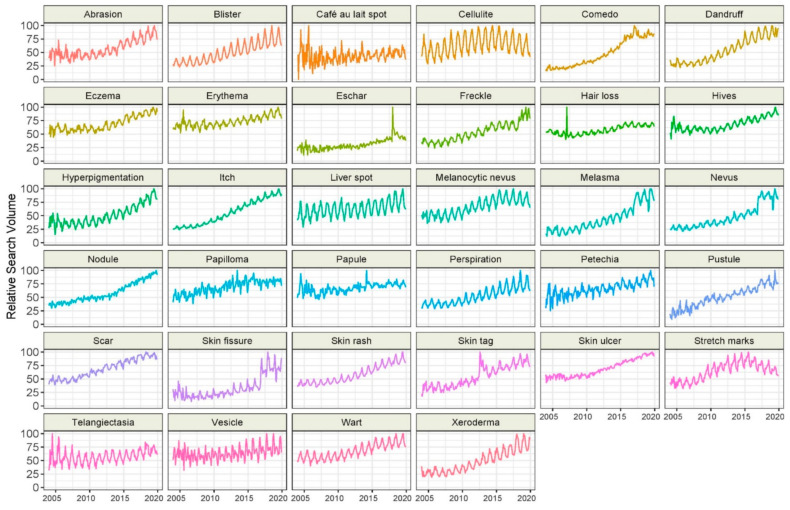
Relative search volume over time of topics related to skin problems. Non-adjusted data.

**Table 1 ijerph-18-02541-t001:** Popularity of topics representing dermatologic complaints in proportion to “Scar” (adjusted data; Relative Search Volume (RSV) over time).

No	Topic	Proportion of RSV in Comparison to Scar
1.	Itch	2.21
2.	Hair loss	1.56
3.	Skin rash	1.38
4.	Perspiration	1.32
5.	Scar	1.00
6.	Wart	0.85
	Pustule	
7.	Blister	0.56
8.	Hives	0.54
9.	Cellulite	0.50
10.	Stretch marks	0.47
11.	Comedo	0.46
	Skin ulcer	0.46
13.	Nevus	0.38
	Nodule	0.38
15.	Dandruff	0.37
16.	Eczema	0.43
	Xeroderma	0.33
18.	Melanocytic nevus	0.32
19.	Erythema	0.28
20.	Freckle	0.26
21.	Papilloma	0.22
22.	Melasma	0.18
23.	Skin tag	0.13
24.	Papule	0.09
25.	Vesicle	0.08
26.	Hyperpigmentation	0.07
	Telangiectasia	0.07
28	Liver spot	0.06
	Petechia	0.06
30.	Abrasion	0.05
	Pustule	0.05
32.	Eschar	0.04
33.	Skin fissure	0.02
34.	Café au lait spot	0.01

**Table 2 ijerph-18-02541-t002:** Time series analysis of non-adjusted topics.

Topic	Seasonal Mann-Kendall Test	Slope [RSV/Year]	TBATS (Seasonality Present, Period [month])	Month with the Highest Seasonal Component [RSV]	Month with the Lowest Seasonal Component [RSV]	Seasonal Component Amplitude [RSV]
Abrasion	tau = 0.70; ***	2.83; ***	YES, 12	June (8.87)	December (−8.66)	17.54
Blister	tau = 0.95; ***	3.37; ***	YES, 12	July (15.47)	December (−9.52)	24.99
Café au lait spot	tau = 0.20; ***	0.47; 0.018	YES, 12	July (12.32)	November (−8.35)	20.67
Cellulite	tau = 0.45; ***	1.15; ***	YES, 12	May (21.61)	December (−22.16)	43.77
Comedo	tau = 0.92; ***	5.15; ***	YES, 12	August (3.27)	October (−2.50)	5.77
Dandruff	tau = 0.92; ***	4.66; ***	YES, 12	January (9.08)	June (−7.60)	16.68
Eczema	tau = 0.75; ***	2.50; ***	YES, 12	May (6.25)	September (−6.69)	12.94
Erythema	tau = 0.73; ***	1.57; ***	YES, 12	July (6.40)	December (−8.37)	14.78
Eschar	tau = 0.70; ***	1.70; ***	YES, 12	June (1.58)	December (−4.87)	6.45
Freckle	tau = 0.86; ***	3.36; ***	YES, 12	June (5.32)	December (−7.75)	13.07
Hair loss	tau = 0.64; ***	1.34; ***	NO, -	-	-	-
Hives	tau = 0.70; ***	2.03; ***	YES, 12	July (5.33)	December (−6.06)	11.39
Hyperpigmentation	tau = 0.80; ***	3.39; ***	YES, 12	June (6.96)	December (−11.05)	18.02
Itch	tau = 0.99; ***	4.83; ***	YES, 12	July (5.32)	December (−3.20)	8.52
Liver spot	tau = 0.72; ***	1.47; ***	YES, 12	July (16.15)	December (−17.20)	33.35
Melanocytic nevus	tau = 0.80; ***	2.73; ***	YES, 12	July (11.93)	December (−12.03)	23.97
Melasma	tau = 0.92; ***	4.62; ***	YES, 12	July (6.20)	December (−8.00)	14.21
Nevus	tau = 0.92; ***	4.17; ***	YES, 12	July (4.87)	November (−5.02)	9.89
Nodule	tau = 0.94; ***	3.78; ***	YES, 12	May (2.31)	December (−5.88)	8.19
Papilloma	tau = 0.73; ***	2.15; ***	YES, 12	October (5.69)	December (−12.64)	18.33
Papule	tau = 0.56; ***	1.09; ***	YES, 12	July (3.65)	December (−5.75)	9.40
Perspiration	tau = 0.90; ***	2.85; ***	YES, 12	July (12.3)	December (−9.71)	22.01
Petechia	tau = 0.76; ***	2.15; ***	YES, 12	May (10.59)	December (−9.04)	19.63
Pustule	tau = 0.90; ***	3.89; ***	YES, 12	July (6.63)	December (−6.4)	13.03
Scar	tau = 0.92; ***	3.63; ***	YES, 12	July (4.26)	December (−5.65)	9.91
Skin fissure	tau = 0.71; ***	3.46; ***	YES, 12	December (12.23)	September (−6.85)	19.07
Skin rash	tau = 0.94; ***	3.34; ***	YES, 12	July (8.35)	December (−5.57)	13.92
Skin tag	tau = 0.83; ***	3.94; ***	YES, 12	August (8.33)	December (−6.14)	14.47
Skin ulcer	tau = 0.86; ***	3.21; ***	YES, 12	October (1.77)	January (−2.73)	4.50
Stretch marks	tau = 0.60; ***	2.52; ***	YES, 12	July (11.51)	December (−11.74)	23.24
Telangiectasia	tau = 0.54; ***	0.96; ***	YES, 12	June (15.05)	December (−15.25)	30.30
Vesicle	tau = 0.52; ***	1.06; ***	YES, 12	October (17.25)	July (−14.24)	31.49
Wart	tau = 0.84; ***	2.42; ***	YES, 12	July (11.78)	December (−11.09)	22.87
Xeroderma	tau = 0.92; ***	4.02; ***	YES, 12	January (10.77)	September (−8.34)	19.11

*** *p* < 0.001; -: If TBATS model does not show seasnality, no period (seasonal cycle) can be calculated.

## Data Availability

The dataset will be available on Mendeley after publication of the paper.

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
