# Peer review of "‘Dr. Google, What Is That on My Skin?’—Internet Searches Related to Skin Problems: Google Trends Data from 2004 to 2019"

_ijerph, 2021, doi:10.3390/ijerph18052541_

Round 1

Reviewer 1 Report

I really enjoyed reading this paper; although a bit unconventional, it is a very interesting paper on the use of Google as a substitute doctor. Very interesting the distribution for each country; only some minor concerns:

I would probably eliminate Europe standalone map, as it is easily accessible from the world map.

I would also eliminate the relative search volume over time maps, as basically all research volumes are increasing, with no exceptions.

page 11 line192-195 "The fourth topic, that most dynamically gained attention, was melasma. The condition affects approximately 1% of the general population and even 9-50% of the high-risk population of the melasma such as pregnant women and those using female hormones [36]." please also cite " Nisticò SP, Tolone M, Zingoni T, Tamburi F, Scali E, Bennardo L, Cannarozzo A new 675 nm Laser Device in the treatment of Melasma: Results of a Prospective Observational Study Photobiomodul Photomed Laser Surg. 2020 Sep;38(9):560-564

Thank you

Reviewer 2 Report

The subject of the paper “Dr. Google, what is that on my skin?’ – Internet searches related to skin problems: Google Trends data from 2004 to 2019” is timely and valuable to the audience of the Information. Researchers did research on users’ interest in „skin” searches with the use of the Google search engine.

Overall, the paper is well structured, reads quite well, and covers the existing literature quite well. The analysis of the data is interesting and well documented. However, to my view, the manuscript is not ready for publication.

The authors used data from 15,5 years, which shows that is increasing for each year, but do not explain what causes the increase (Lines 245-246). Authors assume that relative search volume from Google Trends is adjusted to the number of internet users. However, they do not provide any external statement for this assumption, except their own papers, already published, references 7, 24, 25 - lines 66-68. This assumption (about adjustment to a number of internet users) is wrong, since „Each data point is divided by the total searches of the geography and time range it represents to compare relative popularity. Otherwise, places with the most search volume would always be ranked highest.” Source: https://support.google.com/trends/answer/4365533?hl=en

Data is adjusted between regions, since „Different regions that show the same search interest for a term don't always have the same total search volumes” (same source), but not in one region. I recommend going once again through the methodology, since this cause the unexplained further conclusions about the rising number of searches. For 15,5 years many people started to use the internet, smartphones appeared, very often with Android which uses Google as a standard search engine. The increasing interest is probably correlated with the increased number of internet users and the number of devices connected to the network.

The second major issue is the end of the timeline. The study finishes in the middle of 2019. Why there is no data to the end of 2019 (the title of the paper suggests it)? More interesting would be to extend the data up to 2020, since the COVID-19 pandemic could and for sure did interfere with many clinical and health treatment, thus shakes the medical interest of internet users.

In lines 151 to 155 authors discuss the seasonality and amplitudes. It is not clear if the lowest interest was in February (line 153) or in December (line 154). A similar statement is with the highest interest in August (line 153) or in July (line 154).

In lines 197 - 221 authors discuss the seasonality of the results connected with seasonality in the northern hemisphere. How does it work for countries in the study that are completely in the southern hemisphere, e.g. Argentina, Brazil, or Australia?

In figure 3 there are presented obvious anomalies (Eschar, Hair loss, Dandruff), but authors discuss only one (line 219), with a statement that they noticed this, but do not know what caused it. In this section, these kinds of outliers should be all discussed.

In the introduction section, the authors claim that there are only a few studies using Google data for one specific skin clinical sign in one country (line 58). It is expected that in the discussion section, authors are supposed to compare their results with the previous works, at least with the cited ones in the introduction.

The quality of presentation and text formatting is not sufficient. Each line from the manuscript text is joined with a line number. It makes it difficult to read text fluently, when each line starts with the number merged with the content, without any additional space. I am not aware whether the authors submitted this version or the editorial office made some changes to the text, but it was not easy to follow the content.

Figure 1 is stretched in width - particularly all countries over the world are stretched, except European countries. It doesn’t look like a normal map.

In the supplementary file, there is no figure S1, only a caption for this figure.

Minor language comments:

Line 135: Dataset will be …

Lines 142-144:  - missing dot after sentence

Lines 154: interest was

Of course, I admire the amount of work put into this research. I recommend revise and resubmit this work by taking into account the above comments.

Reviewer 3 Report

The authors should stress that there is probably an age-bias in the presented data as google users are in median younger or tend to become younger. This might explain why perspiration and pustule might be frequently searched through google.

The authors should nevertheless repeat their search with the common names of frequent skin diseases such as acne, atopic dermatitis, psoriasis, hyperhidrosis, warts, scabies, melanoma, squamous cell cancer and basal cell cancer and compare these searches with the results obtained by topics.

Round 2

Reviewer 2 Report

Thank you very much. Almost all of my previous comments were correctly addressed. I think that the manuscript has been significantly improved.

The one thing left is a statement about northern hemisphere (line 213) and my previous comment how does it work for countries in the study that are completely in the southern hemisphere, e.g. Argentina, Brazil, or Australia? I leave it to the authors (or editor decision) if they would like to extend this work on this.

Reviewer 3 Report

I would have preferred that the authors had repeated their search with the common names of frequent skin diseases such as acne, atopic dermatitis, psoriasis, hyperhidrosis, warts, scabies, melanoma, squamous cell cancer and basal cell cancer and had compared these searches with the results obtained by non-medical topics. I understand that this could be considered a completely new study.
